# Environmental Disclosure: Study on Efficiency and Alignment with Environmental Priorities of Spanish Ports

Emma Castelló-Taliani [1], Silvia Giralt Escobar [1] and Fabricia Silva da Rosa [2,*]

1 Department of Economics & Management Sciences, University of Alcalá, 28802 Alcalá de Henares, Spain; emma.castello@uah.es (E.C.-T.); silvia.giralt@uah.es (S.G.E.)
2 Department of Accounting, Federal University of Santa Catarina, Florianópolis, SC 88040-900, Brazil
* Correspondence: fabriciasrosa@hotmail.com

**Abstract:** The purpose of this article is to analyze, in a three-stage research project and from an economic an operational perspective, the relationships between environmental expenses, the improvements achieved in five environmental variables analyzed and efficiency. To achieve these objectives, we analyze sustainability reports and economic data from 24 Spanish ports. The three aforementioned stages of this research are the following: first, the analysis of the sustainability reports to determine the level of information; second, the analysis of the economic and operational efficiency; and, third, the analysis of the alignment with the environmental priorities of the Eco Ports-ESPO (European Sea Ports Organization). The results reveal that (1) the type of traffic does not affect environmental actions; (2) environmental performance (improvements) depends on environmental expenditures; (3) environmental spending and efficiency in port operations are correlated; and (4) environmental spending and port economic efficiency are correlated. The research can contribute to the decision-making process of port managers by revealing that the alignment with the EcoPorts priorities can be important to direct the environmental performance of the ports towards the global interests revealed in this indicator. It also reveals that environmental expenditures and investments may be related to environmental performance and economic and operational efficiency. However, it also reveals that it is important to improve the extent of environmental disclosure to better explain the qualitative and monetary characteristics of each piece of information provided about environmental performance.

**Keywords:** environmental disclosure; green port; environmental indicators; port efficiency; Data Envelopment Analysis (DEA)



## 1. Introduction

Maritime transport is an important type of transport for globalized trade, as it represents approximately 90% of global trade [1]; however, its negative impact on the environment is significant. In this scenario, ports should be taken into account: in order to allow this type of transportation, ports often engage in actions that affect the environment [2], such as air pollution; ballast water discharge; dredging and disposal of dredge materials; and storage, transport and management of hazardous substances [3]. These actions raise the need for an environmental management of ports, which may improve customer satisfaction, corporate image, cost savings and environmental protection [4].

Nonetheless, there is an important paradox, because even though the relevance of an environmental management of ports is well known [5–7], the pressure to ensure competitiveness and globalization [8] represents a managerial stress; that is to say, the need to prioritize the environment sometimes prevents ports from maintaining economic efficiency [9,10].

Given this scenario, the European Sea Ports Organization (ESPO) emphasizes the importance of controlling and monitoring green practices [9]. In particular, it identifies

annually, through the EcoPort (EP) initiative, environmental priorities for European ports with the aim of encouraging environmental management practices. In 2019, EcoPort listed as priorities the following areas: air quality, energy consumption, climate change, noise, relationship with the local community, ship waste, garbage/port waste, land-related port development, dredging operations and water quality.

To date, it is known that a sustainable port environment becomes viable with the active involvement of stakeholders and port operators through the implementation of key sustainable indicators [11]. Studies have shown that economic efficiency achieves its ideal target when ports have a high pro-environmental attitude, implementing proactive green policies [9]. The importance of knowing about external pressures on issues related to environmental management [5,12] and economic efficiency [7,9,10] is also widely recognized.

However, the environmental priorities of ports can change annually [13], since the environmental impact and social pressure on the adopted environmental posture may vary. This variation may be caused both by normative aspects and the pressure of interest groups [6], and by the profile of each port (size, type of cargo and logistics, among others), whose development can be greatly influenced by government policies and regulations, central planning, impact control and market uncertainty [10]. This allows us to identify environmental management priorities (the high-priority areas of common concern in which ports are working) and define the guidance framework and initiatives that the Ports Authorities (PAs) should take [14].

The environmental sustainability of ports is at the same time relevant and complex, as it has been explained: although the environmental management of ports is recognized [5,7], there still exists pressure to ensure competitiveness and globalization [8]. In this context, this article proposes, in a three-stage research project and from an economic and operational perspective, to analyze the relationships between environmental expenses, the improvements achieved in five environmental variables analyzed and the efficiency of ports.

The study is justified from two perspectives: the alignment of ports with global environmental priorities (EcoPorts), and the economic and operational efficiency of the ports. Regarding the first perspective, the literature reveals that sustainable port development encompasses social, economic and environmental factors [11,12]. This sustainable development, including port operations and projects, means having a long-term vision, transparency, legal commitment, information exchange and innovation [11]. However, we still have a challenging scenario that demonstrates the need to have a strategic and holistic vision, since the volume of cargo transported through ports is growing, which worsens environmental impacts. In addition, maritime trade has changed with the increase in modern ports, which, in turn, increases aspects of competitiveness [7]. Sustainability in port systems can be challenging and complex [11], as it integrates organizational aspects and the zone of influence of the port related to the environment, such as the preservation of the coast, morphology and marine biodiversity [9], in addition to economic aspects that allow maintaining competitiveness at the global level [7].

To support managers and society in achieving sustainable development, aspects of environmental management and environmental indicators have helped identify, control and monitor environmental aspects and impacts [15]. The reason is that environmental management is used to identify, measure and manage environmental information [16]. Studies have revealed that, to assist environmental management, indicators and metrics can be employed to control, monitor and verify information. Environmental indicators and port evaluation systems are important for measuring and monitoring issues related to policy, personnel, training, communication, monitoring and environmental auditing [14]. They can reveal the various environmental dimensions, including, but not limited to, water consumption, water quality, carbon footprint, energy consumption and auditing, in addition to enabling the setting of priorities and specific information for port development [4,7,17–20]. It can be useful to define an overview of the position of the port sector

and establish a course of action that allows us to outline future trends [14], which will make the business legitimate for stakeholders [21] while assisting in risk mitigation and environmental protection [19]. In this sense, our study is justified because it allows the analysis of to what extent the environmental sustainability of Spanish ports aligns with the priorities established by EcoPort.

The second justification of the study is the analysis of the economic and operational efficiency of the ports. Recent studies have investigated this complex factor through efficiency analysis instruments, such as Data Envelopment Analysis (DEA). The results of these studies show that the integration between the economic efficiency and environmental performance of ports is important to improve economic value and to sustain the competitiveness of ports, with the environmental performance as the central point [7,9,10,22]. The empirical results of these studies reveal that the DEA was used to analyze the efficiency of the ports and verify a positive relationship between environmental sustainability and economic performance.

Castello-Taliani et al. [7] analyzed the efficiency and environmental information of Spanish ports, and identified which efficient and marginally efficient ports provided the best environmental disclosures. Gobbi et al. [22] analyzed the environmental efficiency of Brazilian ports in relation to plastic waste management using the DEA technique, and discovered that efficiency is not consistent from one year to the next, which can be caused by flawed control practices and procedures and inaccurate information about waste generated and discarded by each port. Castellano et al. [9] found that the critical activities carried out, such as energy-saving programs, actions to reduce air and water pollution and waste management, contribute to improving environmental performance and economic efficiency at the same time. Wang et al. [2] verified in their study on port efficiency using the DEA that the aspects of port cooperation can improve the expected overall production but will lose their advantages with the improvement of the standards of emissions.

Considering the proposed objective and the justifications presented, this article has the following structure: a presentation of the background and research hypotheses, the explanation of the materials and the method, the results and the discussion. At the end of the article, we also include the references used.

## 2. Background and Hypothesis Development

The need for sustainable development, in order to achieve the Sustainable Development Goals (United Nations SDGs—UN) and aiming at the balance between the three pillars of sustainability (economy, environment and society), is articulated in different sectors of society [23]. Among these sectors, Kuznetsov et al. [12] highlight the port sector and the search for "blue wealth" or "wealth created by the various services and assets that the oceans provide", in a global concern with the seas and maritime activities, such as transport and ports. Sustainable port development encompasses social, economic and environmental factors [11,12]. This sustainable development comprises port operations and projects that include long-term vision, transparency, legal commitment, information sharing and innovation [11]. Environmental disclosure supports managers in identifying, measuring and evaluating the aspects and impacts that lead to environmental performance. [15,16]. It allows measuring (i) the degree of environmental disclosure; (ii) the relationship between the variables of environmental performance, transparency and economic performance factors with the degree of environmental disclosure; and (iii) the relationship between the economic performance factor and the degree of environmental disclosure modified by the factors of environmental performance and transparency [23].

Studies have shown that, to assist environmental management and disclosure, indicators and metrics can be considered to control, monitor and evidence information. Environmental indicators and assessment systems for ports are important for measuring and monitoring issues related to environmental policy, personnel, training, communication, monitoring and auditing [14]. They can reveal the various environmental dimensions and information for port development [4,7,17,18,20], useful to prioritize and trace future

trends [14], which helps to mitigate risks and protect the environment [19]. However, the increase in the volume of cargo transported and the modernization of ports reflects at the same time an increase in competitiveness between ports and the environmental impacts of the activity. As a result, it is necessary for managers to have a more strategic and holistic view. [7]. To assist the management of European ports, the ESPO, through EcoPorts, proposes methodologies for controlling and monitoring green practices [9]. The fundamental principle of EcoPorts is to create a level playing field in relation to the environment through cooperation and knowledge sharing between ports. EcoPorts, in addition to publishing annually the top 10 environmental priorities, provides two well-established tools to its members: Self Diagnostic Method (SDM) and Port Environmental Review System (PERS). Updating the top l0 environmental issues is an important exercise, because it identifies the common areas of high-priority concern that ports are working on and defines the framework for guidance and initiatives to be taken by representative bodies [14].

However, the level of environmental expenditures, as well as the port's profile in terms of load, can influence performance [7]. The first and second research hypotheses emerge having in mind that, as we understand it, the level of evidence on environmental priorities can facilitate implementing measures to manage environmental aspects; therefore, hypotheses 1 and 2 are as follows:

**Hypothesis 1 (H1).** *The type of traffic conditions environmental actions.*

**Hypothesis 2 (H2).** *Environmental performance (improvements) depends on environmental expenditures.*

Previous studies have shown that port sustainability is complex, as it involves environmental and economic interests that go beyond the organizational setting. The studies also revealed that environmental aspects have become extremely important for the competitiveness of ports. Consequently, integrating environmental performance and economic efficiency is increasingly important in order to improve the economic value and to maintain said competitiveness, considering environmental performance as the central point [7,9,10,22]. From the context of this research, we understand that environmental performance leads to greater economic efficiency; thus, the third and fourth hypotheses of research emerge:

**Hypothesis 3 (H3).** *Environmental spending and efficiency in port operations are correlated.*

**Hypothesis 4 (H4).** *Environmental spending and port economic efficiency are correlated.*

### 3. Materials and Method

The purpose of this article is to analyze, in a three-stage research project and from an economic and operational perspective, the relationships between environmental expenses, the improvements achieved in 5 environmental variables analyzed and efficiency. The objective is to characterize the following 24 Spanish PAs (out of the 28 existing), based on the result of the analysis of the aforementioned relationships: A Coruña, Almería, Avilés, Bahía de Algeciras, Bahia de Cádiz, Baleares, Barcelona, Bilbao, Cartagena, Castellón, Ceuta, Ferrol-San Cibrao, Gijón, Huelva, Las Palmas, Málaga, Marín y Ría de Pontevedra, Melilla, Motril, S. Cruz de Tenerife, Tarragona, Valencia, Vigo and Vilagarcía. The reference data for the study is 2018. The database was built from data obtained from environmental reports published by ports on their websites. The 4 PAs excluded from the study did not provide information on the variables analyzed (Table 1).

**Table 1.** Variables included in the first stage analysis and scales applied.

| Definition | Scale/items |
|---|---|
| Air Quality—Type of information<br>Waste—Type of information<br>Fuel Consumption—Type of information<br>Water Consumption—Type of information<br>Electricity Consumption—Type of information<br>Noise—Type of information | • No information<br>• Descriptive (D)<br>• Monetary (M)<br>• Quantitative (Q)<br>• Descriptive and Monetary (D&M)<br>• Descriptive and Quantitative (D&Q)<br>• Monetary and Quantitative (M&Q)<br>• Descriptive, Monetary and Quantitative (D, M & Q) |
| Environmental Expenses/Operating Expenses<br>Air Quality Improvement<br>Water Consumption Improvement<br>Waste Improvement<br>Electricity Consumption Improvement<br>Fuel Consumption Improvement<br>ROI | % |
| Main Traffic | • Liquid bulks<br>• Dry bulks<br>• General cargo<br>• Liquid bulks/general cargo<br>• Dry bulks/general cargo<br>• General cargo/passengers |

The purpose is to identify correlations among environmental improvements and operational and economic efficiency of port management, considering environmental expenses, traffic and overhead structure. In the first stage of the research, we analyzed the environmental reports of the 24 PAs which were part of our sample to quantitatively determine the improvements they had achieved in the 5 variables selected for the study: air quality, waste, fuel consumption, water consumption and electricity consumption. We also analyzed the type of information they provided for these variables in their environmental reports. Due to the existence of specific environmental regulations, the main traffic in the PAs was considered.

In the second stage of the study, an economic and operational efficiency analysis was performed. A review of the literature on the efficiency of maritime ports shows that DEA is among the most frequently used quantitative techniques [2,7,22]. In this regard, we refer to the review of the literature conducted by Schøyen and Odeck [24], which documented that, out of 47 articles on port efficiency, 36 used DEA and 11 used Stochastic Frontier Analysis (SFA). In the authors' opinion, this shows that DEA is the most employed tool. Table 2 shows the variables applied in two analyses.

The third stage of the study involved a correlational analysis of the PA environmental reports found in the first stage, and an examination of the results found in the second stage of the economic and operational efficiency analysis. The purpose was to define and characterize environmental groups, based on the information related to environmental improvements and environmental expenditure obtained in the first stage, and compare them with the efficiencies reported by PAs. The data used in this analysis were obtained from the official periodical reports issued by each PA. The environmental evaluation, as well as the DEA economic and operational efficiency analysis, was administered to 24 PAs, which represent 85.7% of Spanish ports of general interest. Therefore, this evaluation maintains its relevance. The Frontier Analyst software was used for the DEA.

**Table 2.** Data Envelopment Analysis (DEA) variables.

| Variables Applied in the DEA Economic Analysis | |
|---|---|
| Inputs | Depreciation and Amortization of Non-current Assets <br> Personnel Expenses <br> Other Operating Expenses |
| Output | Operating Revenue |
| **Variables Applied in the DEA Operational Analysis** | |
| Inputs | Tangible Fixed Assets <br> Number of Employees <br> Total Operating Expenses |
| Outputs | Percentage of Concessional Occupation <br> Thousands of Gross Tones (GT) <br> Metric tons of Goods (Freight Traffic) |

## 4. Results

In the first stage of the investigation, a database was developed including the improvements (+ or −), as a percentage, analyzed by the 24 PAs in 2018. This type of information was collected through the environmental reports (according to the scale set out in Table 1).

In the second stage, we opted for a DEA efficiency analysis with variable returns to scale (Banker, Charnes y Cooper (BCC) model) using the variables previously defined. The model calculates the relative efficiency of each PA, including changes in operational scale to reflect the current reality of the PA comprising the Spanish Ports System. The study performed was designed based on maximum outputs, an essential efficiency factor. The potential actions on the expense structure and non-current assets of PAs are constrained, due to the nature of activity, as the overhead expenses of most PAs are fixed. As noted above, the data used in this analysis were obtained from the official periodical reports issued by each PA. Tables 3 and 4 show the summary of the basic statistics of DEA variables for the period analyzed.

**Table 3.** Basics statistics—inputs.

| | Operating Revenue | Personnel | Amortization | Other Operating Expenses |
|---|---|---|---|---|
| Average | EUR 9,415,979.29 | EUR 13,505,957.75 | EUR 16,899,231.25 | EUR 47,991,864.04 |
| Max. | EUR 32,341,000.00 | EUR 45,588,000.00 | EUR 57,706,000.00 | EUR 180,326,000.00 |
| Min. | EUR 2,779,290.00 | EUR 1,539,858.00 | EUR 2,895,221.00 | EUR 5,056,759.00 |
| Standard deviation (S.D.) | EUR 6,690,252.93 | EUR 11,845,564.87 | EUR 13,895,955.70 | EUR 42,149,184.68 |

**Table 4.** Basics statistics—outputs.

| | Tangible Fixed Assets | Number of Employees | Total Operating Expenses | Percentage of Concessional Occupation | Thousands of GT | TM of Goods (Freight Traffic) |
|---|---|---|---|---|---|---|
| Average | 350,422,371.79 | 204.29 | 37,285,811.83 | 57.23 | 93,704,388.31 | 22,243,226.54 |
| max | 1,435,932,000.00 | 535.00 | 135,635,000.00 | 98.28 | 410,703,181.00 | 102,543,929.32 |
| min | 53,409,264.00 | 62.00 | 7,085,000.00 | 11.65 | 2,698,826.00 | 868,060.50 |
| S.D. | 319,124,662.11 | 115.94 | 30,217,728.81 | 20.38 | 116,675,483.17 | 25,076,961.18 |

The use of DEA with variable returns to scale (to maximize the outputs considered) results in the classification of the efficiency assigned to the different units analyzed, to each of which a value of 0–100% is assigned; that is, it gives the score assigned to efficient and inefficient PAs (see Table 5). The assumptions were applied in DEA, where scores below 100% indicate a relative level of inefficiency.

**Table 5.** DEA scores (2018).

| Unit Name | Economic DEA | | Operational DEA | |
|---|---|---|---|---|
| | Score (%) | RTS | Score (%) | RTS |
| A CORUÑA | 92.76 | 1 | 100.00 | 1 |
| ALMERÍA | 71.4 | −1 | 100.00 | −1 |
| AVILÉS | 90.46 | −1 | 50.10 | 1 |
| BAHÍA DE ALGECIRAS | 82.81 | 1 | 100.00 | 0 |
| BAHIA DE CADIZ | 64.61 | 1 | 66.50 | 1 |
| BALEARES | 100 | 1 | 87.30 | 1 |
| BARCELONA | 100 | 1 | 100.00 | 1 |
| BILBAO | 84.28 | 1 | 100.00 | 1 |
| CARTAGENA | 100 | 1 | 98.70 | −1 |
| CASTELLÓN | 100 | 1 | 100.00 | −1 |
| CEUTA | 100 | −1 | 100.00 | 0 |
| FERROL-SAN CIBRAO | 88.15 | −1 | 84.80 | −1 |
| GIJÓN | 100 | 1 | 64.67 | −1 |
| HUELVA | 91.81 | −1 | 90.10 | 1 |
| LAS PALMAS | 98.66 | 1 | 100.00 | 0 |
| MÁLAGA | 74.41 | −1 | 59.10 | −1 |
| MARÍN Y RÍA DE PONTEVEDRA | 100 | −1 | 74.10 | 1 |
| MELILLA | 78.32 | −1 | 83.60 | −1 |
| MOTRIL | 100 | −1 | 100.00 | −1 |
| S. CRUZ DE TENERIFE | 82.57 | 1 | 100.00 | 0 |
| TARRAGONA | 88.97 | −1 | 100.00 | 0 |
| VALENCIA | 100 | 1 | 100.00 | 1 |
| VIGO | 82.55 | 1 | 58.20 | 1 |
| VILAGARCÍA | 100 | −1 | 100.00 | 0 |

The absolute score was divided among four categories: efficient PAs (score 100%), marginally efficient PAs (≥90%), marginally inefficient PAs (≥80%) and inefficient PAs (<80%), as shown in Table 6.

**Table 6.** PA classification by DEA scores.

```
ECONOMIC DEA

  Code   Meaning                                Frequency    %
  ------ ------------------------------------- ---------- -------
     1   Efficient                                    10    41.66
     2   Marginally Efficient                          3    12.50
     3   Marginally Inefficient                        7    29.16
     4   Inefficient                                   4    16.66
                                              ---------- -------
                          Total                        24   100.00

OPERATIONAL DEA

  Code   Meaning                                Frequency    %
  ------ ------------------------------------- ---------- -------
     1   Efficient                                    13    54.16
     2   Marginally Efficient                          1     4.16
     3   Marginally Inefficient                        4    16.66
     4   Inefficient                                   6    25.00
                                              ---------- -------
                          Total                        24   100.00
```

In the third stage of the study, a cluster analysis was performed. Group or cluster analysis techniques are statistical techniques which help to identify groups that, while otherwise different, are internally homogenous. We used the Johnson algorithm, full chain build-up, for all cluster analyses, in which the distance between two clusters (groups) is taken as the greatest between the elements integrating those groups [25].

The variables considered in the cluster analysis, used to classify PAs based on environmental improvements and expenses information, were the following: environmental

expenses/operating expenses, air quality improvement, water consumption improvement, waste improvement, electricity consumption improvement and fuel consumption improvement. Table 7 contains the dendrogram obtained from this analysis, which shows the classification obtained by the PAs. Table 8 includes the average and standard deviation of the variables, by group, considered in the cluster analysis.

**Table 7.** Dendrogram—PAs classification based on environmental improvements and expenses information.

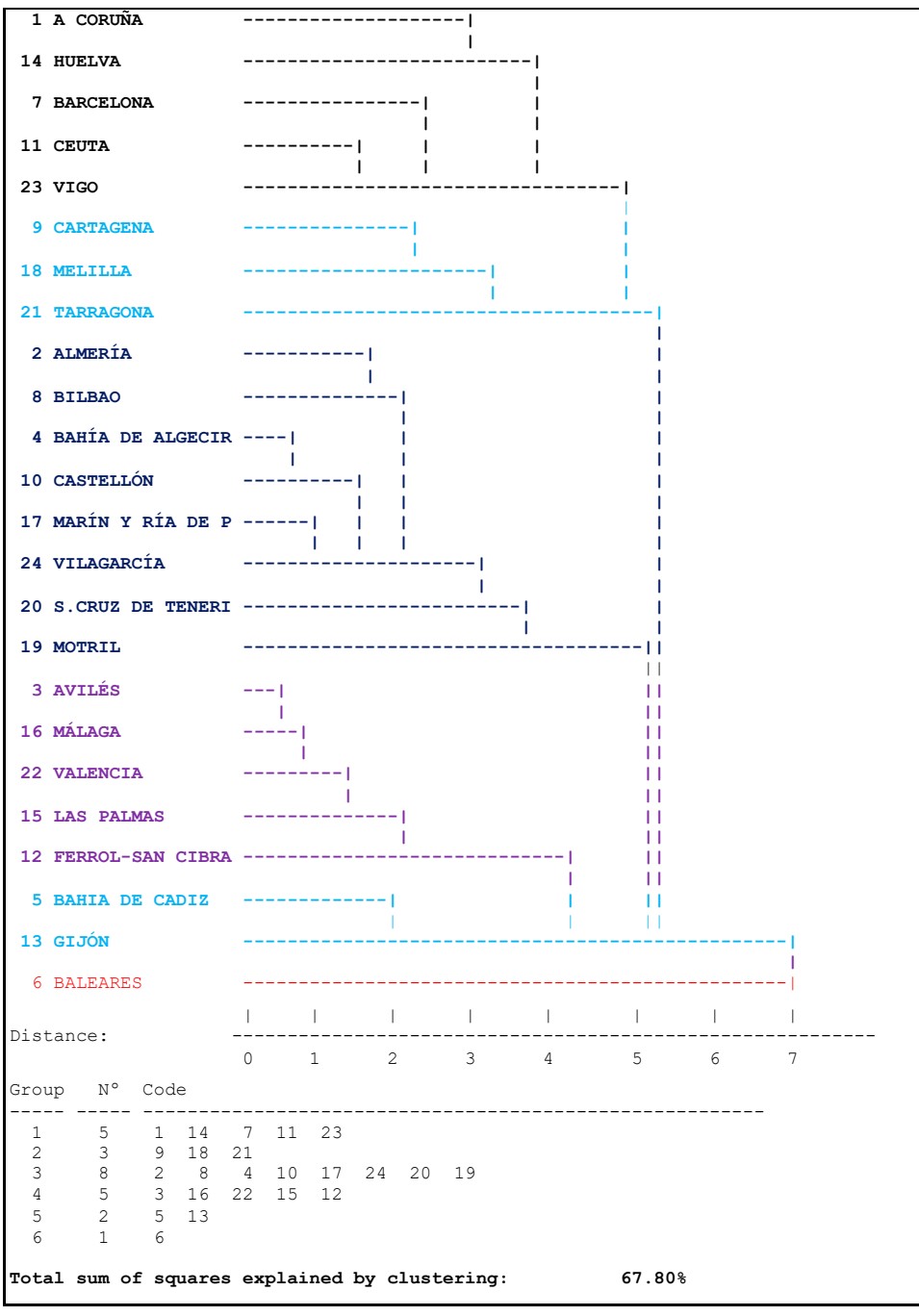

**Table 8.** Average and standard deviation (S.D.) of the variables considered in the cluster analysis.

| | | TOTAL | GROUP 1 | GROUP 2 | GROUP 3 | GROUP 4 | GROUP 5 | GROUP 6 | |
|---|---|---|---|---|---|---|---|---|---|
| | NUMBER: | 24 | 5 | 3 | 8 | 5 | 2 | 1 | |
| Sum of | squared: | 10.08 | 0.74 | 0.75 | 1.45 | 0.16 | 0.15 | 0.00 | |
| Variables: | | | | | | | | | ANOVA / F de Snedecor |
| % enviro | Average: | 0.04 | 0.09 + | 0.01 - | 0.03 | 0.03 | 0.01 | 0.06 | F(5.18) = 4.2624 |
| | S.D.: | 0.04 | 0.04 | 0.00 | 0.02 | 0.02 | 0.01 | 0.00 | (p = 0.0099) |
| air impr | Average: | 0.11 | 0.15 | -0.03 | -0.06 - | -0.02 | 0.92 + | 0.76 | F(5.18) = 12.5028 |
| | S.D.: | 0.34 | 0.17 | 0.13 | 0.21 | 0.08 | 0.08 | 0.00 | (p = 0.0000) |
| water co | Average: | -0.03 | -0.33 | 0.11 | 0.09 | 0.05 | 0.17 + | -0.72 - | F(5.18) = 7.0843 |
| | S.D.: | 0.27 | 0.21 | 0.20 | 0.16 | 0.07 | 0.17 | 0.00 | (p = 0.0008) |
| waste i | Average: | 0.26 | 0.16 | 0.12 | 0.07 - | 0.72 + | 0.30 | 0.24 | F(5.18) = 16.3307 |
| | S.D.: | 0.27 | 0.10 | 0.23 | 0.09 | 0.04 | 0.17 | 0.00 | (p = 0.0000) |
| electric | Average: | 0.01 | -0.06 | -0.18 - | 0.08 | 0.03 | 0.01 | 0.13 + | F(5.18) = 2.4536 |
| | S.D.: | 0.14 | 0.13 | 0.10 | 0.13 | 0.03 | 0.05 | 0.00 | (p = 0.0731) |
| fuels im | Average: | 0.06 | 0.03 | 0.51 | -0.12 - | -0.02 | -0.11 | 1.00 + | F(5.18) = 4.6785 |
| | S.D.: | 0.37 | 0.22 | 0.36 | 0.29 | 0.14 | 0.08 | 0.00 | (p = 0.0065) |

## 5. Discussion

### 5.1. Descriptive Analysis

The variables analyzed to evaluate the environmental actions that have been put in place by the various Spanish Ports Authorities are presented in Table 9.

**Table 9.** Variables used in the analysis.

| |
|---|
| % Environmental Expenses/Operating Expenses |
| Air Quality |
| Waste |
| Fuel Consumption |
| Water Consumption |
| Electricity Consumption |

The variation experienced between 2017 and 2018 was analyzed for each of these variables. The individual values obtained by the different PAs are not detailed in this article; instead, the data were analyzed using a cluster analysis, which allowed for the identification of 6 groups. Table 10 shows the variation rates achieved by each group in each of the variables mentioned above.

If these rates are analyzed, a well-defined behavior can be observed in the six groups. Group 6 stands out, as it shows an above-average environmental expenditure, which has translated into important above-average improvements in air quality, fuel consumption and electricity consumption. Despite achieving a major improvement in waste, it is below average. On the contrary, Group 1 assembles the Ports Authorities that allocate an above-average environmental expenditure and managed to improve both air quality and waste. Group 5 only allocates 1% of its operation expenditure to environmental expenditure, although it achieves improvement ratios that are above the corresponding averages in air quality, waste and water consumption. Group 3 and 4 allocate 3% of their operational expenditure to the environment achieving an above-average improvement in water consumption and electricity consumption. With regard to Group 2, its 1% environmental expenditure has allowed it to improve more than the average in fuel consumption and electricity consumption.

It is worth noting that the groups that have allocated a higher percentage of their expenditure to the environment have not managed to improve in water consumption. In the case of Group 6, which is an insular Port Authority, this can be caused by its high needs for this element and its reduced capacity to economize.

**Table 10.** Variation percentages for environmental variables.

|  | Group 1 | Group 2 | Group 3 | Group 4 | Group 5 | Group 6 | Average |
|---|---|---|---|---|---|---|---|
| % environmental expenses | 9% | 1% | 3% | 3% | 1% | 6% | 4% |
| Air quality | 15% | –3% | –6% | –2% | 92% | 76% | 11% |
| Waste | 16% | 12% | 7% | 72% | 30% | 24% | 26% |
| Fuels consumption | 3% | 51% | –12% | –2% | –11% | 100% | 6% |
| Water consumption | –33% | 11% | 9% | 5% | 17% | –72% | –3% |
| Electricity consumption | –6% | –18% | 8% | 3% | 1% | 13% | 1% |

If the variation rates presented in Table 8 are compared to the variations shown by the Ports Authorities (between 2017 and 2018) regarding the size of the ships that have circulated (measured in Gross Tones (GT) Thousands) and the tons of goods, a more discernible behavior can be observed. Figure 1 represents the variations observed in the activity levels of the six groups identified in the Ports Authorities between 2017 and 2018.

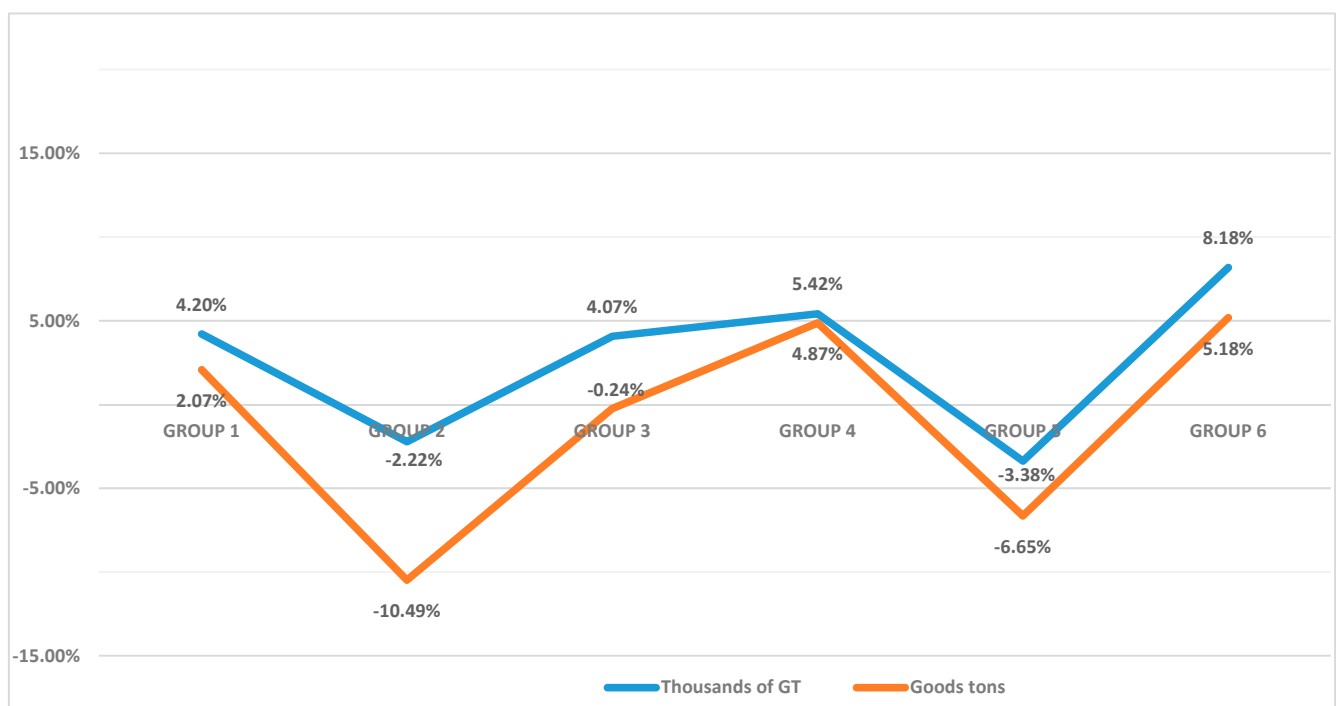

**Figure 1.** Variations in the activity levels of the cluster groups.

Group 6 is the one that displays an increase in the port activity compared to the others, both in absolute numbers, regarding the average variations experienced by the G.T., and the tons of goods. As for this, we can assert that, despite the increase in the port activity, the improvements in the environmental variables are above the average, except for water consumption, which experiences the worst variation rate, as it did not achieve savings in the use of this resource.

Group 4 also experiences an economic activity variation rate above the corresponding averages, categorizing its environmental actions as very positive, except for the deterioration experienced in air quality, caused by a larger number of ship movements, and fuel consumption. It is worth noting that in this group, two of the biggest Ports Authorities in containerized general cargo and one of the biggest ports in solid bulk are included, which results in more air pollution.

In Group 1, an increase in G.T. and tons of goods above the corresponding averages is noted. This group shows deterioration in water consumption and electricity consumption. In this group, some of the Ports Authorities that manage liquid bulk cargo can be found, which can be the cause preventing a savings increase in water consumption. However, the increase in electricity consumption is caused by the absence of measures to improve the efficiency when consuming this resource.

Group 3 shows a bigger movement of ships and a lower number of tons of goods. This could be the origin of air quality deterioration, although the lack of energy efficiency is obvious with regard to fuel consumption, as it deteriorated over 2018. Groups 2 and 5 show a negative economic activity variation rate and display an uneven behavior in air quality and electricity consumption, which worsen in Group 2, even though fuel consumption worsens in Group 5.

Considering the information on the variables used in the group definition listed in Table 5 (previously analyzed) and the characterization of groups by traffic, reflected in Figure 2, data could be obtained that allowed us to contrast Hypotheses 1 (the type of traffic conditions environmental actions) and 2 (environmental performance improvements depend on environmental expenditures).

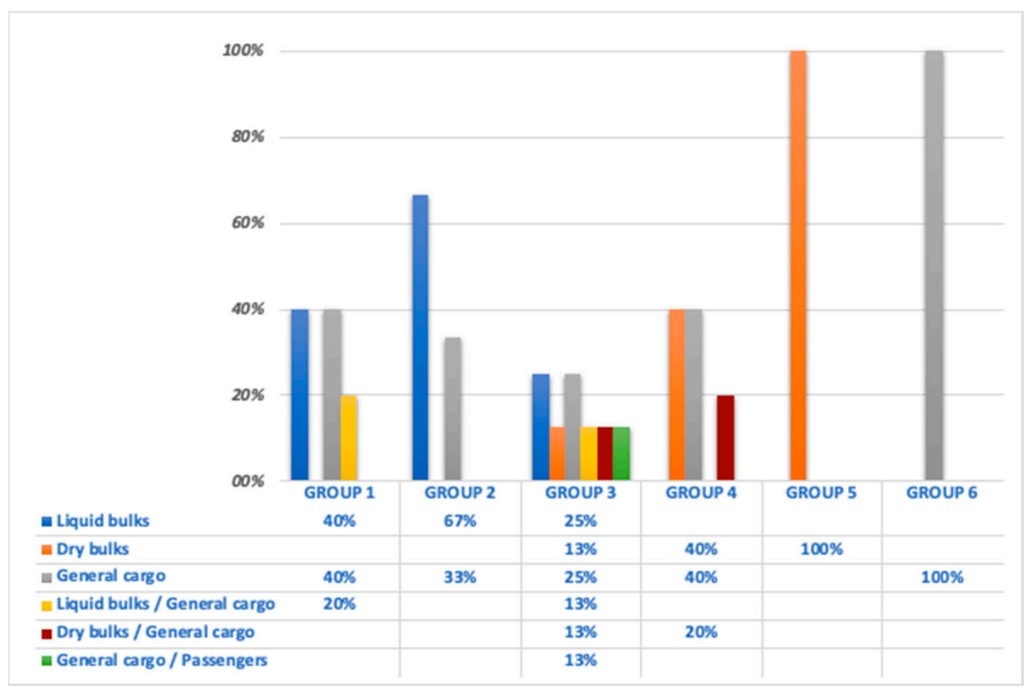

**Figure 2.** Traffic by group.

It was verified in the port sector that the type of cargo can be decisive for the type and level of environmental impact; that is, cargo considered dangerous (such as liquid cargo) is more likely to have environmental impacts. However, the results regarding the disclosure of environmental performance do not allow us to confirm the first hypothesis, because the groups with this load profile were not the ones that obtained the highest level of evidence. Based on the information presented, the H1 hypothesis cannot be confirmed, whereas h2 can.

The characterization of the groups obtained, based on the two DEA carried out and considering environmental variables (see Tables 11–13), shows that Group 1, which has the highest environmental expenditure, is also the one that obtains the best efficiency results calculated from the economic and operational variables. The behavior of Groups 3 and 4, with an environmental expenditure of around 3%, is similar in terms of economic efficiency, but not in terms of operational efficiency. The lowest environmental expenditure groups, 2 and 5, are groups that show a higher level of operational and economic inefficiency,

although both have members who are efficient from an economic perspective. Group 6, with a single PA, is an efficient group from an economic perspective, and marginally inefficient from an operational perspective (DEA score = 87.30%).

**Table 11.** Group by operational variables DEA.

```
                                          cluster
                 -----------------------------------------------------------------------
                     GROUP 1     GROUP 2     GROUP 3     GROUP 4     GROUP 5     GROUP 6
                 TOTAL
                 ----------- ----------- ----------- ----------- ----------- ----------- -----------
DEAOP_C          Frec    %   Frec    %   Frec    %   Frec    %   Frec    %   Frec    %   Frec    %
---------------- ---- ------ ---- ------ ---- ------ ---- ------ ---- ------ ---- ------ ---- ------
1 Efficient       13  54.17   3  60.00   1  33.33   7  87.50   2  40.00   0   0.00   0   0.00
2 Marginally Eff   2   8.33   1  20.00   1  33.33   0   0.00   0   0.00   0   0.00   0   0.00
  icient
3 Marginally Ine   3  12.50   0   0.00   1  33.33   0   0.00   1  20.00   0   0.00   1 100.00
  fficient
4 Inefficient      6  25.00   1  20.00   0   0.00   1  12.50   2  40.00   2 100.00   0   0.00
                 ---- ------ ---- ------ ---- ------ ---- ------ ---- ------ ---- ------ ---- ------
     TOTAL        24  (24)    5   (5)    3   (3)    8   (8)    5   (5)    2   (2)    1   (1)

     Ji squared 15 degree of freedom = 23.8897   (p = 0.0670)
```

**Table 12.** Group by economic variables DEA.

```
                 -----------------------------------------------------------------------
                     GROUP 1     GROUP 2     GROUP 3     GROUP 4     GROUP 5     GROUP 6
                 TOTAL
                 ----------- ----------- ----------- ----------- ----------- ----------- -----------
DEAECO_C         Frec    %   Frec    %   Frec    %   Frec    %   Frec    %   Frec    %   Frec    %
---------------- ---- ------ ---- ------ ---- ------ ---- ------ ---- ------ ---- ------ ---- ------
1 Efficient       10  41.67   2  40.00   1  33.33   4  50.00   1  20.00   1  50.00   1 100.00
2 Marginally Eff   4  16.67   2  40.00   0   0.00   0   0.00   2  40.00   0   0.00   0   0.00
  icient
3 Marginally Ine   6  25.00   1  20.00   1  33.33   3  37.50   1  20.00   0   0.00   0   0.00
  fficient
4 Inefficient      4  16.67   0   0.00   1  33.33   1  12.50   1  20.00   1  50.00   0   0.00
                 ---- ------ ---- ------ ---- ------ ---- ------ ---- ------ ---- ------ ---- ------
     TOTAL        24  (24)    5   (5)    3   (3)    8   (8)    5   (5)    2   (2)    1   (1)

     Ji squared 15 degree of freedom= 11.5833   (p = 0.7103)
```

Based on the information presented, Hypotheses H3 (environmental spending and efficiency in port operations are correlated) and H4 (environmental spending and port economic efficiency are correlated) can be confirmed.

**Table 13.** Group by efficiencies and environmental expenses.

| % Environmental Expenses (Group Average) | Group 1 | Group 2 | Group 3 | Group 4 | Group 5 | Group 6 |
|---|---|---|---|---|---|---|
| | **Higher Average (9%)** | **Lowest Average (1%)** | **3%** | **3%** | **1%** | **6%** |
| DEA OPERATIONAL | 60% efficient | 33.33% efficient | 87.5% efficient | 40% efficient | 100% inefficient | 100% marginally inefficient |
| | 20% marginally efficient | 33.33% marginally inefficient | | 20% marginally inefficient | | |
| | 20% inefficient | 33.333% inefficient | 12.5% efficient | 40% inefficient | | |
| DEA ECONOMIC | 40% efficient | 33.33% efficient | 50% efficient | 20% efficient | 50% efficient | 100% efficient |
| | 40% marginally efficient | 33.33% marginally inefficient | 37.5% marginally inefficient | 40% marginally efficient | 50% inefficient | |
| | 20% marginally inefficient | 33.333% efficient | 12.5% inefficient | 20% marginally inefficient | | |
| | | | | 20% inefficient | | |

### 5.2. Environmental Disclosure and Alignment with Environmental Priorities

The analysis of environmental disclosure is performed from the elements considered a priority in EcoPorts2018. Although the EcoPorts system is complex in terms of environmental management, the individual data of the analyzed ports are not for public access. The information is consolidated and presented in the annual report without detailed information on the port and performance of each element. Even with this limitation, the report allows the priorities established by European ports to be followed.

In 2018, according to EcoPorts2018, the priorities of European ports were the following: (1) air quality, (2) energy consumption, (3) noise, (4) relationship with local community, (5) ship waste, (6) port development, (7) climate change, (8) water quality, (9) dredging operations and (10) garbage/port waste.

This information comes from the self-assessment that ports perform, which is later evaluated by EcoPorts. However, such information is not public. Nevertheless, the sustainability reports released by the ports do not necessarily highlight the information provided to EcoPorts. In this sense, this research identified that the information in the reports is not aligned with the elements of EcoPorts, as it presents general information on environmental performance. The elements highlighted in the reports are air, water, noise, waste, energy and fuel, as indicated in Table 14.

According to Table 12, some important issues may be observed, such as the alignment between the environmental priorities listed by EcoPorts, taken from the responses of the ports to this organization system, and the sustainability reports issued by the ports to the public. The current investigation found that reports present information about 6 of the 10 priorities, emphasizing the three main ones of the EcoPorts: air, energy and noise. This may represent an alignment with EcoPorts and, at the same time, a search for legitimacy, as pointed out by the previous literature [21].

It is also possible to verify that Groups 1, 5 and 6 show greater amplitude in terms of dissemination, regarding information related to descriptive and quantitative data. However, this may represent a limitation of the scope and quality of information because, according to the previous literature, the information disclosed on environmental aspects is expected not only to describe the situation of each element (air, energy, noise, waste, fuel and water) but also to provide quantitative and monetary information. This demonstrates that the ports analyzed have not linked environmental issues to economic ones. This may limit information, which may also hinder the stakeholders' understanding of the financial effort made to address the environmental issues of ports, as pointed out by the previous literature [7].

Environmental sustainability in port systems integrates aspects internal and external to the ports (organizational and zone of influence) [9], in addition to economics [7]; therefore,

the level of disclosure can be ambiguous [5,7], as environmental sustainability in ports is challenging and complex [11].

**Table 14.** Port traffic and environmental performance.

| Type of Environmental Information | GROUP 1 | GROUP 2 | GROUP 3 | GROUP 4 | GROUP 5 | GROUP 6 |
|---|---|---|---|---|---|---|
| Traffics | Liquid Bulks General Cargo Liquid Bulks/General Cargo | Liquid Bulks General Cargo | All Types of Traffic Defined | Dry Bulks General Cargo Dry Bulks/General Cargo | Dry Bulks | General Cargo |
| Air (EcoPort1) | 80% group information D&Q | 1/3 group D, 1/3 D&Q and 1/3 Q | 62.5% group D&Q | 60% group D&Q | 100% group D&Q | 100% group D&Q |
| Energy (EcoPort2) | 80% group information D&Q | 66.66% group Q | 50% group Q and 50% D&Q | 60% group D&Q | 100% group D&Q | 100% group D&Q |
| Noise (EcoPort3) | 60% group information D&Q | 100% group D | 62.5 % group D&Q | 60% group no information | 50% group no information and 50% D&Q | 100% group D&Q |
| Waste (EcoPort5) | 80% group information D&Q | 66.66% group D&Q | 50% group D&Q | 60% group D&Q | 100% group D&Q | 100% group D&Q |
| Fuel (*EcoPort7) | 66% group Q | 1/3 group D, 1/3 D&Q y 1/3 Q | 50% group Q y 50% D&Q | 40% group D y 50% D&Q | 100% group D&Q | 100% group D&Q |
| Water (EcoPort8) | 80% group D&Q | 66.66% group Q | 62.5 % group D&Q | 60% group D&Q | 100% group D&Q | 100% group D&Q |

Additionally, it was verified that Group 1, where all ports manage dangerous net cargo, included ports with the highest environmental expenditure. This may demonstrate that there is no disclosure or that there are expenses necessary to manage performance or environmental impacts. However, the information about the expenditure is limited (without detailing the type of expenditure), which does not allow us to confirm the first hypothesis of this research, as shown in Table 15.

**Table 15.** Evolution of the 2018 information in relation to the previous year and environmental performance.

| Environmental Expenditures (E.S) | Group 1 | Group 2 | Group 3 | Group 4 | Group 5 | Group 6 |
|---|---|---|---|---|---|---|
| % E.S. | 9% | 1% | 3% | 3% | 1% | 6% |
| Air quality (EcoPort1) | 15% | −3% | −6% | −2% | 92% | 76% |
| Electricity (EcoPort2) | −6% | −18% | 8% | 3% | 1% | 13% |
| Fuel (EcoPort7) | 3% | 51% | −12% | −2% | −11% | 100% |
| Water (EcoPort8) | −33% | 11% | 9% | 5% | 17% | −72% |
| Waste (Ecoport10) | 16% | 12% | 7% | 72% | 30% | 24% |

## 6. Conclusions, Limitations and Future Research

### 6.1. Conclusions

The purpose of this article is to analyze, in a three-stage research project and from an economic and operational perspective, the relationships between environmental expenses, the improvements achieved in five environmental variables analyzed and efficiency.

The objective is to characterize the following 24 Spanish PAs (out of the 28 existing), based on the result of the analysis of the aforementioned relationships: A Coruña, Almería, Avilés, Bahía de Algeciras, Bahia de Cádiz, Baleares, Barcelona, Bilbao, Cartagena, Castellón, Ceuta, Ferrol-San Cibrao, Gijón, Huelva, Las Palmas, Málaga, Marín y Ría de Pontevedra, Melilla, Motril, S. Cruz de Tenerife, Tarragona, Valencia, Vigo and Vilagarcía. The reference data for the study are from 2018. The purpose is to identify correlations among environmental improvements and operational and economic efficiency of port management, while considering environmental expenses, traffic and overhead structure.

In the port sector, it was verified that the type of cargo can be a determinant of the type and level of environmental impact, and that the results regarding the disclosure of environmental performance do not allow us to confirm the first hypothesis, because the

groups with this load profile were not the ones that obtained the highest level of evidence. The characterization, based on the two DEAs completed, of the groups obtained considering environmental variables (see Tables 11–14) shows that environmental expenditures can influence the environmental performance in Groups 3, 4 and 6, but the same cannot be said for the other groups, which demonstrates that the idea of more expenses resulting in better performance is not conclusive, because other issues, such as the type of load, can influence this performance.

The study reveals a relevant alignment of the information provided by ports and environmental priorities listed by EcoPorts, taken from the responses of the ports to the EP system and the sustainability reports issued by the ports to the public. The current research found that the reports present information about 6 of the 10 priorities, emphasizing the three main priorities of the EP: air, energy and noise.

The research contributes theoretically to the theme of environmental disclosure, since it allows us (i) to identify the specific environmental disclosure variables for ports, (ii) to expand knowledge about environmental priorities for ports, (iii) to analyze environmental performance and (iv) to analyze operational and economic efficiency. Empirically, it can assist in making management decisions on the environmental performance of ports, environmental expenditures and investments, environmental performance and economic and operational efficiency.

*6.2. Limitations and Future Research*

We were also able to find limitations within the study. First, it was not possible to identify the extension of the elements of environmental performance, since most of the information is descriptive and quantitative. Second, although we expected the type of cargo transported to be related to the environmental actions developed in the ports, it was not possible to verify this hypothesis in the ports analyzed. It is also important to report that the port sustainability reports underwent substantial changes from year 2017, in addition to the fact that there are no more recent publications of these reports (2019 and 2020), which may represent a lack of transparency and timeliness. In addition, ports and EcoPorts do not disclose the responses to the questionnaire administered to ports that serves as a basis for choosing environmental priorities. We understand that the transparency of this information also allows us to better understand the context of management, disclosure and environmental efficiency of ports.

For future research, it is considered relevant to analyze the role of coercive and voluntary elements regarding environmental dissemination and alignment with environmental priorities of international organizations. In addition, our study sheds light on future studies to analyze the interaction between priority environmental aspects and port competitiveness aiming at global sustainability, such as integration with the sustainable development goals (SDGs) agreed between 193 States Members of the United Nations (UN).

**Author Contributions:** Conceptualization, E.C-T., S.G.E. and F.S.d.R.; methodology, E.C-T., S.G.E. and F.S.d.R.; software, E.C-T., and S.G.E.; validation, E.C-T., and S.G.E.; formal analysis, E.C-T.; investigation, E.C-T., S.G.E. and F.S.d.R.; resources, E.C-T., S.G.E. and F.S.d.R.; data curation, E.C-T., and S.G.E.; writing—original draft preparation, E.C-T., S.G.E. and F.S.d.R.; writing—review and editing, E.C-T., S.G.E. and F.S.d.R.; visualization, E.C-T., S.G.E. and F.S.d.R.; supervision, E.C-T.; project administration, E.C-T.; funding acquisition, E.C-T., S.G.E. and F.S.d.R. All authors have read and agreed to the published version of the manuscript.

**Funding:** This research was funded by Conselho Nacional de Desenvolvimento Científico e Tecnológico—CNPq, grant number 306073/2018-8" and "University of Alcalá".

**Data Availability Statement:** The data presented in this study are available on request from the corresponding author.

**Conflicts of Interest:** The authors declare no conflict of interest.

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
