# Peer review of "Environmental Disclosure: Study on Efficiency and Alignment with Environmental Priorities of Spanish Ports"

_sustainability, doi:10.3390/su13041791_

Round 1
Reviewer 1 Report
The paper is interesting and well-written, while, at the same fits well with the Journal's scope. However, before it can be published, the following major amendments should be addressed:
- The paper's abstract should be enriched by including a reference to the paper's justification as well as a refrence to the paper's contribution.
- The authors state that they have used data concerning 24 out of 28 Spanish ports? They should justify why they didn't use data for all the ports and explain possible bias in the research results with the ommiting of the 4 ports.
- A complete justification based on the relevant literature is needed concerning the examined variables.
- The research hypohteses should be further supported by adding more literature. Furthermore, it would be better that this section should be added as a new section below Introduction (e.g. "Hypotheses Developement").
- The DEA model used should be furhter defined and explained. The authors should explain why they used this model in contrast to others.
- Figure 6 should be rearranged or replaced by another more interpretable one
- The authors should try to recognize their paper's novelty and contribution in the conclusions section.
- Last, a new section with possible limitations, caveats and future resarch directions should be added.
The following minor amendments should be addressed as well:
- The Journal's reference style should be used.
- The paper should be carefully proofread so as some grammar and language mistakes are corrected.
Author Response
Reviewer 1
“Environmental Disclosure: Study on Efficiency and Alignment with Environmental Priorities of Spanish Ports”
Response to reviewer 1 report Sustainability – January 2021
Thank you very much for your positive comments and suggestions. The major concerns of the English language, background, hypothesis, discussion, conclusion and future research. Bearing these improvements in mind, in the following pages we will try to explain how we considered and responded to your concerns and suggestions. We highlight the changes to our manuscript by using green text.
R1Q01: The paper's abstract should be enriched by including a reference to the paper's justification as well as a refrence to the paper's contribution
A: We reviewed the summary to better present the research and its results.
The purpose of this article is to analyze, in a three-stage research, the relationships between environmental expenses, the improvements achieved in 5 environmental variables analyzed and efficiency, from an economic and operational perspective. To achieve these objectives, we analyze sustainability reports and economic data from 24 Spanish ports. The stages of this research are analyzing the sustainability reports to determine the level of information, analyzing the economic and operational efficiency, and analyzing the alignment with the environmental priorities of the Eco Ports-ESPO (European Sea Ports Organization). The results reveal that (1) the type of traffic does not condition environmental actions; (2) environmental performance (improvements) depends on environmental expenditures; (3) environmental spending and efficiency in port operations are correlated; and (4) environmental spending and port economic efficiency are correlated. The research can contribute to the decision making of port managers by revealing that the alignment with the EcoPorts priorities can be important to direct the environmental performance of the ports to the global interests revealed in this indicator. It also reveals that environmental expenditures and investments may be related to environmental performance, and economic and operational efficiency. However, it reveals that it is important to improve the extent of environmental disclosure to explain the qualitative and monetary characteristics of each information provided about environmental performance.
R1Q02: The authors state that they have used data concerning 24 out of 28 Spanish ports? They should justify why they didn't use data for all the ports and explain possible bias in the research results with the ommiting of the 4 ports.
A: We included the justification in item 3
.
The reference data for the study is 2018. The database was built from data obtained from environmental reports published by ports on their websites. The 4 PAs excluded from the study did not provide information on the variables analyzed.
R1Q03: A complete justification based on the relevant literature is needed concerning the examined variables
A: We appreciate the reviewer's suggestion as it improves the article. Thus, we made an additional effort and opened the topic of background and hypotheses to present the theoretical support of the research. We also removed the hypotheses from the methods section for this new topic 2 (Background and Hypothesis Development)..
R1Q4: The research hypohteses should be further supported by adding more literature. Furthermore, it would be better that this section should be added as a new section below Introduction (e.g. "Hypotheses Developement").
A: We appreciate the reviewer's suggestion as it improves the article. Thus, we made an additional effort and opened the topic of background and hypotheses to present the theoretical support of the research. We also removed the hypotheses from the methods section for this new topic 2 (Background and Hypothesis Development).
R1Q5: The DEA model used should be furhter defined and explained. The authors should explain why they used this model in contrast to others.
A: We present a justification for using the DEA model in item 3 (Materials and Method)
In this regard, we refer to the review of the literature conducted by Schøyen & Odeck [24], which documented that, out of 47 articles on port efficiency, 36 used DEA and 11 used Stochastic Frontier Analysis (SFA). In the authors’ opinion, this shows that DEA is the most employed tool.
R1Q6: Figure 6 should be rearranged or replaced by another more interpretable one
A: We split the table in two to improve the presentation of the data.
R1Q7: The authors should try to recognize their paper's novelty and contribution in the conclusions section.
A: We appreciate your contribution to improve the conclusion, we seek to better present the news and contribution of the conclusion session, as per the paragraph below.
The research theoretically contributes to the theme of environmental disclosure when it allows (i) to identify the specific environmental disclosure variables for ports, (ii) and to expand knowledge about environmental priorities for ports, (iii) and to analyze environmental performance, (iv) to analyze operational efficiency. and economical. Empirically it can assist management decisions on environmental performance of ports, environmental expenditures and investments, environmental performance, and economic and operational efficiency.
R1Q8: Last, a new section with possible limitations, caveats and future resarch directions should be added.
A: We have included a section within the conclusion to present limitations and future research, as per the paragraph below.
6.2. Limitations and future research
We were also able to find limitations within the study. Firstly, it was not possible to identify the extension of the elements of environmental performance, since most of the information is descriptive and quantitative. Secondly, although we expected the type of cargo transported to be related to the environmental actions developed in the ports, it was not possible to verify this hypothesis in the ports analyzed. It is also important to report that the port sustainability reports underwent substantial changes in the years 2017 onwards, in addition to the fact that there are no more recent publications of these reports (2019 and 2020), which may represent a lack of transparency and timeliness. In addition, ports and EcoPorts do not disclose the responses to the questionnaire made to ports that serves as a basis for choosing environmental priorities. We understand that the transparency of this information also allows us to better understand the context of management, discloisure and environmental efficiency of ports.
For future research, it is considered relevant to analyze the role of coercive and voluntary elements regarding environmental dissemination and alignment with environmental priorities of international organizations. In addition, our study sheds light on future studies to analyze the interaction between priority environmental aspects and port competitiveness aiming at global sustainability, such as, for example, the integration with the sustainable development gools (SDGs) agreed between 193 States Members of the United Nations (UN).
R1Q9: The Journal's reference style should be used.
A: We have carried out the review for the Journal's reference style
R1Q10: The paper should be carefully proofread so as some grammar and language mistakes are corrected
A: We have carried out anew vision of English
Reviewer 2 Report
Dear Authors,
thank you for your interesting manuscript.
It contains interesting information dealing with the efficiency and alignment of Spanish ports to environmental priorities.
The literature has to be improved: you can find relevant manuscript on dry port in many MDPI journals (for example, Sustainability and Logistics).
The conclusion have to bee improved to better tie together the other elements of the manuscript.
Author Response
Reviewer 2
“Environmental Disclosure: Study on Efficiency and Alignment with Environmental Priorities of Spanish Ports”
Response to reviewer 2 report Sustainability – January 2021
Thank you very much for your positive comments and suggestions. The major concerns of the background and conclusion and future research. Bearing these improvements in mind, in the following pages we will try to explain how we considered and responded to your concerns and suggestions. We highlight the changes to our manuscript by using green text.
R1Q01: thank you for your interesting manuscript, It contains interesting information dealing with the efficiency and alignment of Spanish ports to environmental priorities.
A: Thanks.
R1Q02: The literature has to be improved: you can find relevant manuscript on dry port in many MDPI journals (for example, Sustainability and Logistics).
.
A: We appreciate the reviewer's suggestion as it improves the article. Thus, we made an additional effort and opened the topic of background and hypotheses to present the theoretical support of the research. We also removed the hypotheses from the methods section for this new topic 2 (Background and Hypothesis Development)
R1Q03: The conclusion have to bee improved to better tie together the other elements of the manuscript.
A: We appreciate your contribution to improve the conclusion, we seek to better present the news and contribution of the conclusion session, as per the paragraphs below.
The research theoretically contributes to the theme of environmental disclosure when it allows (i) to identify the specific environmental disclosure variables for ports, (ii) and to expand knowledge about environmental priorities for ports, (iii) and to analyze environmental performance, (iv) to analyze operational efficiency. and economical. Empirically it can assist management decisions on environmental performance of ports, environmental expenditures and investments, environmental performance, and economic and operational efficiency.
6.2. Limitations and future research
We were also able to find limitations within the study. Firstly, it was not possible to identify the extension of the elements of environmental performance, since most of the information is descriptive and quantitative. Secondly, although we expected the type of cargo transported to be related to the environmental actions developed in the ports, it was not possible to verify this hypothesis in the ports analyzed. It is also important to report that the port sustainability reports underwent substantial changes in the years 2017 onwards, in addition to the fact that there are no more recent publications of these reports (2019 and 2020), which may represent a lack of transparency and timeliness. In addition, ports and EcoPorts do not disclose the responses to the questionnaire made to ports that serves as a basis for choosing environmental priorities. We understand that the transparency of this information also allows us to better understand the context of management, discloisure and environmental efficiency of ports.
For future research, it is considered relevant to analyze the role of coercive and voluntary elements regarding environmental dissemination and alignment with environmental priorities of international organizations. In addition, our study sheds light on future studies to analyze the interaction between priority environmental aspects and port competitiveness aiming at global sustainability, such as, for example, the integration with the sustainable development gools (SDGs) agreed between 193 States Members of the United Nations (UN).
Reviewer 3 Report
Dear authors,
In the framework of sustainable development, green ports are a very important issue. The paper is well-prepared. The methods are adequate. The hypotheses were chosen appropriately. The results and discussion confirm / do not confirm the chosen hypotheses. English should be revised as there are few grammar mistakes.
Below are some minor inaccuracies and suggestions for improvement:
- Try to rewrite the abstract to more accurately characterize your paper.
- Line 177-178 – use also italic style (same as h3 and h4)
- Table 3 - please explain „S.D.“ in the table so that even the disinterested reader understands the given values.
- Line 193 – analysed
- Figure 2 - is quite unreadable, try to improve its quality
- Line 520 - one extra space in a line
Congrats for your research!
Author Response
Reviewer 3
“Environmental Disclosure: Study on Efficiency and Alignment with Environmental Priorities of Spanish Ports”
Response to reviewer 3 report Sustainability – January 2021
Thank you very much for your positive comments and suggestions. The major concerns of the abstract, english and arguments and discussion of findings. Bearing these improvements in mind, in the following pages we will try to explain how we considered and responded to your concerns and suggestions. We highlight the changes to our manuscript by using green text.:
R2Q01: In the framework of sustainable development, green ports are a very important issue. The paper is well-prepared. The methods are adequate. The hypotheses were chosen appropriately. The results and discussion confirm / do not confirm the chosen hypotheses. English should be revised as there are few grammar mistakes.
A: Thank you so much for your review that help us improve the article. We seek to meet your requests to improve results and English.
R2Q02: Try to rewrite the abstract to more accurately characterize your paper.
A: We reviewed the summary to better present the research and its results.
The purpose of this article is to analyze, in a three-stage research, the relationships between environmental expenses, the improvements achieved in 5 environmental variables analyzed and efficiency, from an economic and operational perspective. To achieve these objectives, we analyze sustainability reports and economic data from 24 Spanish ports. The stages of this research are analyzing the sustainability reports to determine the level of information, analyzing the economic and operational efficiency, and analyzing the alignment with the environmental priorities of the Eco Ports-ESPO (European Sea Ports Organization). The results reveal that (1) the type of traffic does not condition environmental actions; (2) environmental performance (improvements) depends on environmental expenditures; (3) environmental spending and efficiency in port operations are correlated; and (4) environmental spending and port economic efficiency are correlated. The research can contribute to the decision making of port managers by revealing that the alignment with the EcoPorts priorities can be important to direct the environmental performance of the ports to the global interests revealed in this indicator. It also reveals that environmental expenditures and investments may be related to environmental performance, and economic and operational efficiency. However, it reveals that it is important to improve the extent of environmental disclosure to explain the qualitative and monetary characteristics of each information provided about environmental performance.
R2Q03: Line 177-178 – use also italic style (same as h3 and h4)
A: We reviewed the style of the h3 and h4 hypotheses.
R2Q04: Table 3 - please explain „S.D.“ in the table so that even the disinterested reader understands the given values.
A: At the end of table three we insert the explanation for the expression S.D.
R2Q07: Line 193 – analysed
A: Adjusted
R2Q08: Figure 2 - is quite unreadable, try to improve its quality
A: Adjusted
R2Q09: Line 520 - one extra space in a line
A: Adjusted
R2Q10: Congrats for your research!
A: Thanks
Round 2
Reviewer 1 Report
The authors have addressed all the comments adequately. Thus, the paper is now eligible for publication.
Reviewer 2 Report
Dear Authors,
I have appreciate your effort and now, in my opinion, your manuscript is ready to be published. Congratulation!